Perspective

# Rational strategies for improving the efficiency of design and discovery of nanomedicines

Xiaoting Shan [1,2], Ying Cai[1,2,3], Binyu Zhu[1,2], Lingli Zhou[1], Xujie Sun[1,2], Xiaoxuan Xu[1,2], Qi Yin [1,2], Dangge Wang[4] ✉ & Yaping Li [1,2,3,5] ✉

The rise of rational strategies in nanomedicine development, such as high-throughput methods and computer-aided techniques, has led to a shift in the design and discovery patterns of nanomedicines from a trial-and-error mode to a rational mode. This transition facilitates the enhancement of efficiency in the preclinical discovery pipeline of nanomaterials, particularly in improving the hit rate of nanomaterials and the optimization efficiency of promising candidates. Herein, we describe a directed evolution mode of nanomedicines driven by data to accelerate the discovery of nanomaterials with high delivery efficiency. Computer-aided design strategies are introduced in detail as one of the cutting-edge directions for the development of nanomedicines. Ultimately, we look forward to expanding the tools for the rational design and discovery of nanomaterials using multidisciplinary approaches. Rational design strategies may potentially boost the delivery efficiency of next-generation nanomedicines.

Rational strategies, such as high-throughput strategies and computer-aided techniques, are increasingly widespread in nanomedicine development to meet the demand for designing nanoparticles efficiently, considering the diversity and complexity of nanomedicines and improving the hit rate of effective candidates in the vast chemical structure space[1,2]. The transition of the design pattern from a trial-and-error mode to a rational mode is particularly prominent in the process of lipid discovery and has been gradually extended to the design of various nanomaterials. For instance, ionizable lipids are core components of nucleic acid drug delivery systems in clinical applications, and small changes in the chemical properties of lipids can greatly affect biological functions[3,4]. The identification of ionized lipids as core components is a bottleneck in the development of lipid nanoparticles. However, discovering and optimizing promising lipid candidates through human-centered trial-and-error strategies is a time-consuming process, and some chemical structures may be difficult

to discover by human intuition. For example, ionizable lipid DLin-MC3-DMA (MC3), the key component of the Food and Drug Administration (FDA)-approved lipid nanoparticle loaded with siRNA (Onpattro), has undergone a long journey of structural optimization (from 2005 to 2012) to improve delivery efficiency and safety[5–7]. Subsequently, combinatorial chemistry was used for the high-throughput synthesis of lipid libraries to generate structurally diverse libraries for screening effective candidates. However, physical libraries based on combinatorial synthesis often cover limited structural spaces and may miss some promising candidates[8]. Fortunately, the rise of computer-aided high-throughput technologies (e.g., artificial intelligence and virtual screening) is reshaping the process of nanomaterial design and discovery, leading to a shift in the development mode of nanomedicine towards a data-driven and rational pattern. Specifically, computer-aided nanomedicine design and discovery strategies have demonstrated satisfactory potential to expand the structural space of

[1]State Key Laboratory of Drug Research & Center of Pharmaceutics, Shanghai Institute of Materia Medica, Chinese Academy of Sciences, Shanghai 201203, China. [2]University of Chinese Academy of Sciences, No. 19A Yuquan Road, Beijing 100049, China. [3]Yantai Key Laboratory of Nanomedicine & Advanced Preparations, Yantai Institute of Materia Medica, Yantai, Shandong 264000, China. [4]Precision Research Center for Refractory Diseases, Shanghai General Hospital, Shanghai Jiao Tong University School of Medicine, Shanghai 201260, China. [5]Shandong Laboratory of Yantai Drug Discovery, Bohai Rim Advanced Research Institute for Drug Discovery, Yantai, Shandong 264117, China. ✉e-mail: dg_wang@sjtu.edu.cn; ypli@simm.ac.cn

nanoparticle building blocks to improve the likelihood of identifying effective candidates, accelerate the screening process, speed up the mapping of structure-activity relationships, and elucidate nano-bio interactions.

Herein, we provide an overview of the current progress in data-driven rational strategies for improving the efficiency of the design and discovery of nanomedicines in preclinical pipelines. In the rational design and discovery mode, nanomedicines may evolve in the same way as biological systems (Fig. 1) through diversification, screening, and optimization of nanomaterial properties, thus selecting the desired candidates[9,10]. Rational strategies such as machine learning, virtual screening, and barcode technology are expected to accelerate important steps in the directed evolution of nanomedicines, especially diversification and screening. It should be emphasized that although the application of computer-aided methods in nanoparticle design has been the focus of this review, computer-based methods can only play a role in assisting the design and accelerating the efficiency of material discovery. Experimental knowledge and verification are irreplaceable.

## Expanding the design space of nanomedicines

The diversification of nanoparticles is the primary step in the directed evolution. Combination synthesis and computer-aided virtual screening will enrich the structural diversity of the composition of nanoparticles, thereby expanding their design space. Expanding the size and diversity of molecular libraries is important for obtaining high hit rates for lead discovery[2]. Modular designs facilitate combinatorial synthesis of a large number of structurally diverse compounds from a small number of starting chemicals[11]. Designing units with standardized joints will contribute to diversified nanoparticle libraries[12]. Chemical reactions with mild reaction conditions and high reaction efficiency, such as Michael addition, amine epoxide ring-opening reaction, and click reaction, can be used to synthesize the building blocks of nanocarriers in a high-throughput manner[13]. Isocyanates can be used as both electrophiles and nucleophiles and can perform

multistep reactions under mild conditions. In one study, using isocyanate isocyanate-mediated one-pot reaction, a three-dimensional combinatorial synthesis library containing 1080 lipids was constructed[14]. The three-dimensional combined synthesis strategy efficiently obtained structurally diverse lipids and did not require toxic catalysts, solvent exchange, or protection/deprotection steps. Combined with microfluidic technologies, more than 1000 lipid nanoparticles were synthesized and evaluated, and the top candidate liposomes enabled the effective delivery of mRNA and activation of immunity. In a follow-up study, compared to three-component reactions, high-throughput platforms based on four-component reactions based on the Ugi reaction have been used to increase chemical dimensions to create a wider array of ionizable lipid candidates, increasing the diversity and structural flexibility of ionizable lipids, and thereby accelerating the discovery of effective ionizable lipids for formation of lipid nanoparticles for precise mRNA delivery[4]. In another study, modularized synthesis of diverse lipids (572 kinds) was efficiently achieved via ring-opening reactions between alkylated dioxaphospholane oxide molecule amines[15]. This modular synthesis strategy enabled precise control of the hydrophobic tail, zwitterion number, and organ selectivity of the liposomes. Combinatorial and high-throughput techniques make it possible to develop nanomedicines in a way similar to the advanced drug discovery of small molecules[16].

To fabricate nanoparticle libraries and study the relationships between multiple physicochemical properties and delivery performance, bottom-up or top-down synthesis methods that realize the rapid, precise, and reproducible synthesis of nanostructures with distinct features (e.g., specified shape and size) are needed[16]. Nucleic acid assemblies via the bottom-up self-assembly of DNA or RNA allow the formation of nanostructures with controllable size and shape[17]. The diversity of nucleic acid sequences and base pairing between natural or artificial nucleic acids allows the design of programmable molecular interactions and performs a broad design space[17]. Nucleic acid assemblies can be fabricated with high spatial accuracy using tile

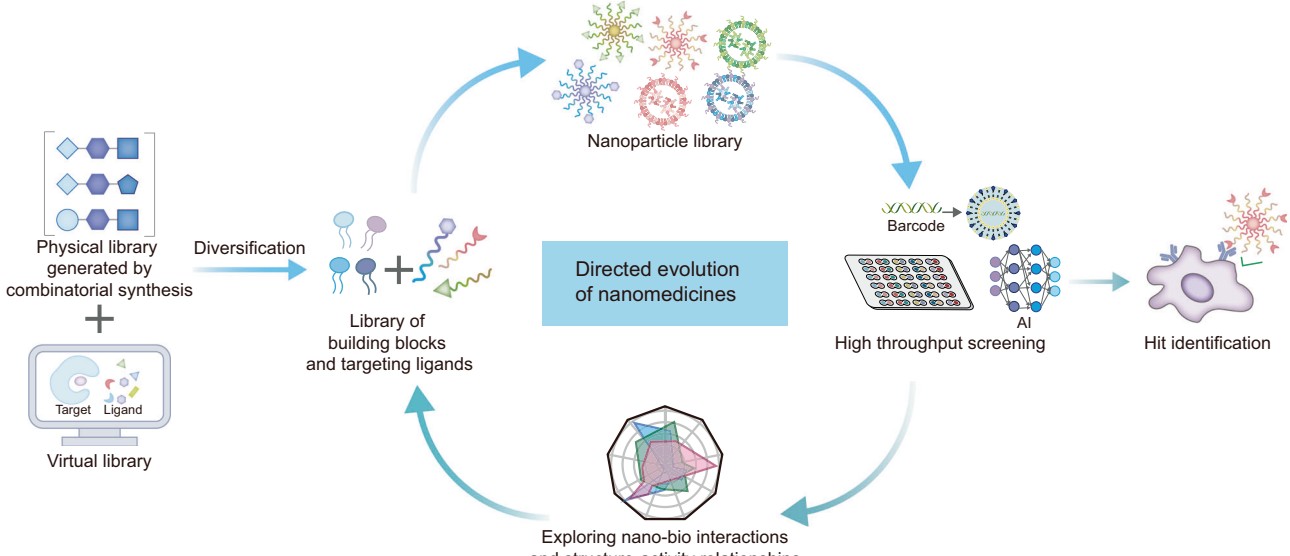

**Fig. 1 | Schematic illustration of directed evolution for improving the efficiency of nanomedicine design and discovery.** Diversification and selection are important processes in directed evolution that improve the hit rate for the discovery of effective candidates. The combination of virtual and physical libraries can realize the diversification of structures for nanoparticle fabrication. The construction of virtual compound libraries combined with computer-aided strategies is beneficial for enriching accessible chemical structures, thus expanding the design space of nanomaterials, which may be difficult to achieve with tangible physical libraries. Modular design and combinatorial synthesis can efficiently generate diverse chemical structures that can be used as building blocks to form nanoparticle libraries. DNA or peptide barcodes are promising strategies for improving the in vivo screening throughput for nanomedicines. Machine learning can accelerate high-throughput screening of the chemical space of nanoparticles, mine the structure-activity relationships, and promote iterative optimization. In addition, the key factors affecting nano-bio interactions identified by data-driven computer-aided strategies can provide feedback to guide the rational design of nanomedicines.

assembly, origami techniques, rolling circle amplification, and nanoparticle-templated assembly[17]. A library of nucleic acid cages with diverse shapes and backbones has been constructed for screening optimal tumor-specific carriers[18]. Microfluidic technology enables the high-speed self-assembly of nanoparticles with narrow size distributions, adjustable physicochemical properties, and greater reproducibility[16]. Particle replication in non-wetting templates (PRINT) is a versatile top-down method for the synthesis of nanoparticles. In the PRINT process, the non-wettability of materials and surfaces limits the liquid precursor inside the mold to generate isolated particles, allowing precise control of particle size, shape, and composition[19]. These synthetic techniques may enable the development of nanomedicines similar to high-throughput screening-based drug discovery patterns and phenotypic drug discovery modes[16,20].

In addition to tangible physical libraries, the construction and screening of virtual libraries can help enrich the accessible chemical structures for the diversification of nanomaterials. Although computer-aided strategies have been widely used to accelerate the discovery of small-molecule drugs, computational chemistry (e.g., molecular docking to screen virtual libraries) has not yet been widely used in designing nanomedicines[21]. One possible reason is that existing computer-aided structure-based drug discovery strategies are commonly used to design and screen small-molecule drugs that bind to specific targets (usually proteins). In addition, compared with small molecules, nanomedicines have a larger size and require a longer simulation timescale, which may overburden existing physics-based calculation methods when computing the formation process of large numbers of nanoparticles and the interactions of nanoparticles with biological macromolecules[22]. Simplifying computational objects may allow virtual screening to be integrated into the nanomaterial discovery pipeline to enrich the diversity of nanoparticles and accelerate the discovery of effective candidates. Specifically, treating the building blocks of nanoparticles as computational objects, rather than entire nanoparticles, is expected to reduce the computational burden. In one study, to optimize the loading capability of polymers, molecular docking was used to screen the small-molecule library, calculate the interaction forces between small-molecule compounds, and screen the compounds with the lowest docking energy to determine the best drug-binding module of polymer terminal modification[23]. The virtual screening strategy based on molecular docking screened the compound library by calculating the interactions between drug-binding modules in polymers and small-molecule drugs, rather than directly stimulating the dynamic process of co-assembly of drug-polymer complexes. This could reduce the demand for calculations and bypass the obstacle of computational chemistry in the rational design of polymers. Compared with physical libraries constructed by wet experiments, the construction of virtual libraries combined with virtual screening may have unique advantages because it allows the screening of a wider range of compound libraries and costs less[2], helping to expand the chemical space of nanomaterials and discover chemical combinations that are difficult to explore using empirical methods. In another study, an extended virtual library containing 40000 lipids was formed by combining different amine heads, linkers, and lipid tails. By screening diverse lipid structures, it is interesting to find that the linkers of high-performing ionizable lipids contain a bulky adamantyl group, which is different from the structure of classical ionizable lipids[4]. Molecular docking and molecular dynamics simulation (MD) provide information on intermolecular interactions, providing promising strategies for high-throughput, rapid, and economical screening of building blocks in nanoparticle formulations[23,24]. In addition, MD allows the simulation of intermolecular interactions in near-real environments (e.g., explicit solvent conditions). MD can provide rich computational chemistry information, but this requires a large amount of computational resources. MD based on coarse-grain force fields may be more suitable for high-

throughput screening because coarse-grained MD can typically achieve acceleration of three orders of magnitude compared to the all-atom force field[25]. In one study, coarse-grained MD was used to explore the large chemical space (8000 possible tripeptides) formed by the combination of 20 amino acids and to rapidly screen candidates capable of self-assembly to form nanostructures[24]. In addition, the molecular information provided by MD revealed the self-assembly rule of peptides, providing guidance for peptide-based nanomedicines. The integration of machine learning, MD, and high-throughput experiments has the potential to expand the chemical structure, improve the throughput of screening, and discover self-assembled compound combinations that are difficult to discover using human intuition. In one study, from 2.1 million pairs of possible drug-excipient combinations formed by 788 drug candidates and 2686 approved excipients, self-assembled nanoparticles with high drug-loading capacities were identified by a platform combining machine learning, MD, and high-throughput experimental technology[26]. These drug-excipient pairs can form stable self-assemblies based on solvent exchange without chemical synthesis. In addition to utilizing existing compound libraries, chemical entity discovery techniques in medicinal chemistry (e.g., pharmacophore hybridization) can be used to enrich the diversity of building blocks for nanoscale self-assembly. In one study, using the principles of pharmacophore hybridization and molecular self-assembly, One-component chemical entities capable of forming nanoscale assemblies were developed by hybridizing lysosomal detergents (MSDH) and autophagy inhibitors (Lys05), effectively inducing lysosomal destruction and improving tumor therapy[27]. This strategy of seamless linkage between small molecules and nanotechnology could accelerate the development of nanomedicine by leveraging its interdisciplinary advantages. We envision that pharmacophore hybridization strategies combined with virtual screening and molecular dynamics simulation have the potential to enhance the structural diversity of building components of self-assembled nanoparticles and increase the throughput of screening. Specifically, virtual excipient libraries can be further expanded through pharmacophore hybridization or structure-activity relationships, and compounds with binding abilities or self-assembly trends can be screened through virtual screening. Promising candidates are tested and enter the next round of optimization. Furthermore, integrating advanced generative models (e.g., diffusion networks) may be a promising strategy for generating diverse chemical structures[8].

Diverse modification ligands generated by current directed evolution methods, such as biopanning methods for peptides and systematic evolution of ligands by exponential enrichment approaches for aptamers[28], can enrich the diversity of the nanoparticle library, from which candidates with high delivery efficiency may be found. In one study, phage display techniques were used to screen variable heavy domains of heavy-chain libraries against human epidermal growth factor receptor 2 (HER2) to modify immunoliposomes[29]. The chimeric nanobody-integrated immunoliposomes loaded with the drug effectively increased cytotoxicity against cancer cell lines overexpressing HER2. Other encoding and display technologies, such as DNA-encoded library technology (DEL) and solid-phase screening of one bead one compound[30], are expected to enrich ligand structures for nanoparticle modification in a high-throughput manner and discover effective candidates.

## Increasing the screening throughput of nanomedicines

Improving the screening throughput of nanoparticles helps explore the vast chemical spaces required to obtain effective candidates. DNA or peptide barcodes are powerful tools for increasing the screening throughput of nanomedicines in vitro and in vivo and can link the physicochemical properties of nanoparticles with delivery efficiency[31,32]. Specifically, nanoparticles or cancer cells can be labeled

with different short DNA sequences (barcodes) and detected using sequencing technology after administration[33–35]. A large number of nanoparticle libraries can be tested simultaneously on experimental animals to quantify the properties (e.g., targeting ability) of nanoparticles in vivo and evaluate the responses of different species and subgroups of cells after treatments[34,36]. The high-throughput screening method based on barcodes comprehensively reveals the connections between the properties of nanoparticles and intracellular internalization, which compensates for the limitations of testing nanoparticles in a single cell line[33]. Despite this progress, there is still a need to improve the sensitivity, accuracy, and throughput of screening strategies for nanoparticles[35].

Artificial intelligence (AI) models utilize a data-driven method of learning that does not rely on explicit formulas and does not require too much prior knowledge or artificial assumptions[37]. Compared with physics-based computational methods, AI models may be more advantageous for predicting the performance of nanomedicines, assisting high-throughput screening of libraries and mining rules to guide the design (Box 1). The iterative optimization between the measured experimental data and AI models can make the design and discovery process of nanoparticles more rational. Specifically, experimental data can be used as input to establish AI models. The AI model can further guide the design, screening, and optimization of nanomedicine to generate data to optimize models. In one study, to alleviate the burden of labor-intensive and time-consuming lipid library screening, machine learning was integrated to accelerate the screening process. The data from the initial library of 584 ionizable lipids were used to train machine-learning models, which were then deployed to probe an expansive virtual library of 40,000 lipids to identify effective lipids[4]. Through AI-based prescreening of a virtual library, the number of promising candidates that need to be synthesized and evaluated in a wet-lab experiment is reduced[8]. In another study, quantitative structure-activity relationship models were established to quantitatively predict the self-assembly of sulfated indocyanine-based nanoparticles and were used to screen drug libraries[38]. A decision tree model was established to screen for drugs that could form nanoparticles with IR783 from the drug library (up to 5653 drugs). The combination of machine learning and wet-lab experiments helps to increase the screening throughput and explore a wider range of chemical spaces, and thus, has the potential to discover effective nanomedicines that may be difficult to discover by human intuition. By combining AI, combinatorial synthesis, and high-throughput screening, it is beneficial to accelerate the mapping of composition-structure-property relationships and change the development mode of nanomedicines[39,40].

## Elucidating nano-bio interactions and reversely engineering nanomedicines

After finding the optimal nanoparticles suitable for certain applications, reverse engineering can be used to discover the mechanisms underlying (nano-bio interactions), and nanoparticles can be further optimized based on nano-bio interactions[41]. Multiple techniques such as MD, machine learning, and multi-omics are expected to accelerate the investigation of nano-bio interactions between nanomedicines and biological barriers. Clarification of these nano-bio interactions can also be utilized to guide the rational design of nanomedicines. Taking the protein corona as an example, it is one of the most important biological barriers encountered during intravenous administration[42]. The nanoparticle-protein interface is critical to the in vivo fate and is thought to affect the clearance and distribution of nanoparticles. Studies have shown that the protein corona and subsequent interactions with highly expressed receptors in tissues dominate the organ-specific distribution of nanoparticles, beyond the previously assumed surface charges of nanoparticles[43,44]. These findings encourage researchers to consider the possibility of other factors that have not been highlighted before. MD and molecular docking reveal the hydrogen bonding, salt bridge, interaction energy, and other interactions between nanoparticles and proteins, providing a powerful tool for the dynamic study of protein adsorption behaviors and interface interact mechanisms[45]. An interesting study extended machine learning algorithms for protein-protein interactions to inorganic nanoparticle-protein complexes by analyzing chemical, geometrical, and graph-theoretical descriptors[46]. These algorithms predict the interaction sites between nanoparticles and proteins and may provide guidance for designing protein-nanoparticle assemblies using rich knowledge of protein-protein interactions. Machine learning algorithms allow researchers to make full use of protein properties in public databases to predict key proteins that form the protein corona, explore the characteristics of the protein corona, and provide preliminary tools for rapid prescreening of candidate proteins when designing nanomedicines[42,47]. Furthermore, machine learning shows the potential to predict the composition of protein corona and the subsequent recognition of nanoparticles by different types of cells[48]. The prediction of nanoparticle-protein interactions and their correlation with the following distribution behaviors in vivo through rational strategies would be a promising and meaningful area of future work. A deeper understanding of nanoparticle-protein interface interactions may turn the disadvantages of the protein corona into advantages by controlling the regulation of the protein corona. For instance, the specific targeting of nanoparticles may be achieved by regulating the adsorption mode of the protein corona[42,43]. Leveraging the nature of binding, we envision that structure-based virtual screening and high-throughput strategies based on intermolecular binding (e.g., DEL)[49] may be applied to the discovery of ligands for nanoparticle modification to improve delivery efficiency. Specifically, these high-throughput techniques based on intermolecular binding are expected to discover ligands that can bind to specific proteins in the protein corona, which can act as target heads to modify nanoparticles, and thus regulate the adsorption pattern of the protein corona for organ-specific targeting (Fig. 2).

In addition to the protein corona, computer-aided techniques have shown great potential in deepening the systematic understanding of various barriers to nanomedicines, such as vascular permeability and internalization[33,50]. To study the heterogeneity of tumor blood vessels quantitatively, a high-throughput single-vessel analysis strategy based on nanoprobes and image-segmentation-based machine learning was developed[50]. Using machine learning, the permeability of heterogeneous tumor blood vessels was quantified and classified. Based on the discovered tumor vascular features and transport mechanisms, genetically tailored protein nanoparticles have been developed to improve the transendothelial transport in low-permeability tumors. This provides a good example of how a deep understanding of nano-bio interactions can guide the rational design of nanomedicines. Machine learning models have the potential to identify the properties of nanoparticles that show important effects on biological functions, explore the influences of key properties, and mine design rules. To identify key factors affecting the internalization of nanoparticles, high-throughput screening, machine learning, and multi-omics were used[33]. The primary factor determining the internalization of nanoparticles by cancer cells was the core component of the nanoparticles rather than the surface materials or modifications, as previously thought. In addition, based on machine learning and protein-protein interaction networks, the lysosomal transporter SLC46A3 has been identified as a negative regulatory factor and predictive biomarker for the uptake of lipid nanoparticles. These rational strategies show great potential to unearth the key factors influencing nano-bio interactions, which could be difficult to discover via empirical methods.

Notably, the optimal engineering design criteria vary at different steps of drug delivery. For instance, the optimal size

## BOX 1

# Applying artificial intelligence to nanomedicines

AI models have the potential to address nonlinear processes, analyze high-dimensional variables, and mine potential relationships[72]. The general construction of machine learning (ML) models includes defining tasks, data collection, selection of appropriate data representation methods (feature engineering), selection of models, and evaluation and optimization of the models. High-quality and sufficient datasets as inputs are vital for training machine learning models[72]. The data used for model training can be generated by computations (such as MD and calculations of the physicochemical properties of compounds)[26,73] produced by wet experiments (e.g., data from high-throughput screening or omics)[10,74,75], or their combinations[22]. In addition, data mining from published materials and imaging information extraction can also provide input data for model training[76–78]. However, the heterogeneity of resources and lack of a unified standard are obstacles to data mining for nanomedicines[72,79]. The establishment of uniform standards and databases will facilitate quantitative comparison, meta-analysis, and computer modeling of the biological effects of nanomedicine, as well as gain knowledge of nanoparticle design patterns[31,80,81]. It is also essential to establish effective descriptors for nanomedicine. The validity and relevance of descriptors affect the accuracy of modeling[72]. Nanomedicines are more complex than small-molecule compounds. Therefore, nanomedicine-specific descriptors need to be developed and optimized. It was found that chemical structure descriptors alone were not sufficient to evaluate the loading ability of polymers for specific compounds[21]. The possible sources of descriptors representing the properties of nanoparticles and the challenges in developing descriptors for nanoparticles can be further referred to in the review[82]. Currently, there is no consensus on which AI algorithms can be applied to specific types of datasets. The utilization of bagging or stacking methods allows the creation of ensembles of different algorithms and may facilitate the development of more robust models[72]. Notably, to reduce the limitation of data amount on model establishment, some algorithms, such as reinforcement learning, transfer learning, and meta-learning models, have been developed to obtain robust models from a low-data environment[72,83].

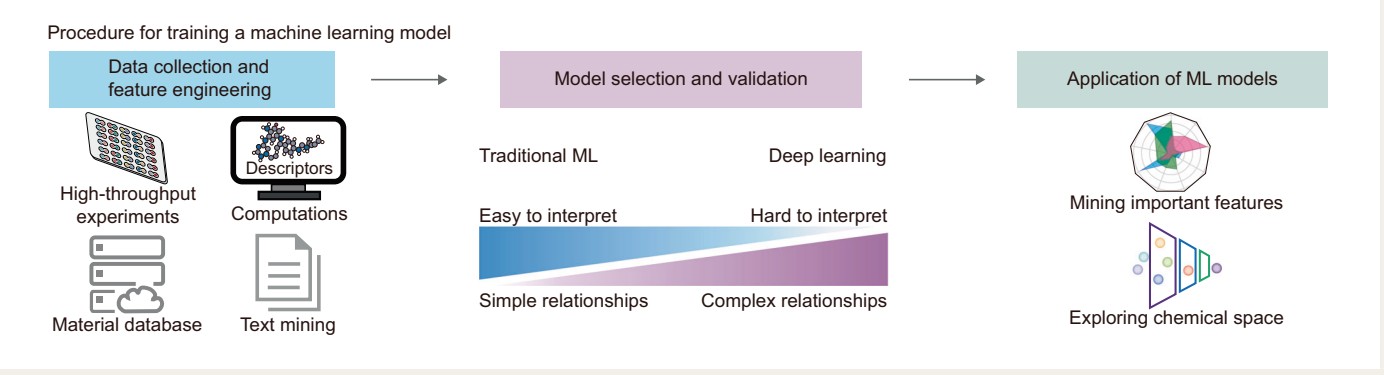

Procedure for training a machine learning model

conducive to long circulation may differ from the size required for deep penetration in tumors[51]. Computer-aided strategies help to investigate nano-bio interactions and structure-activity relationships and then reversibly contribute to the discovery of materials that may resolve design conflicts. For example, nanoparticles consisting of zwitterionic poly(N-oxide) have been found to exhibit both prolonged circulation and penetration into tumors[52]. The adsorption on cell membranes and non-stickiness towards proteins of zwitterionic poly(N-oxide) are considered atypical because polyzwitterions are usually both cell- and protein-resistant. It is expected that after multiple rounds of diversification, screening, optimization, and reverse engineering, it will be possible to determine the design rules and evolve nanoparticles capable of efficient delivery.

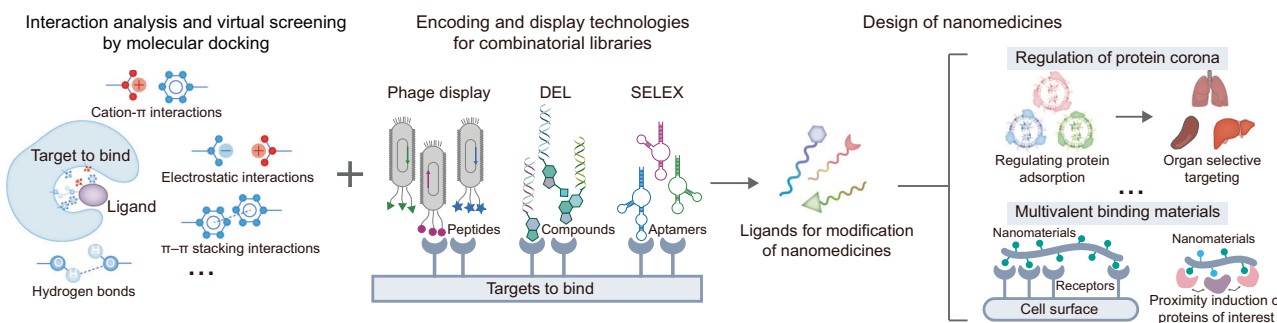

**Fig. 2 | Elucidating nano-bio interactions and reverse-engineering of nanomedicines.** A deeper understanding of nanoparticle-protein interface interactions may facilitate the rational design of nanomedicines. Leveraging the nature of binding, structure-based virtual screening and high-throughput strategies based on intermolecular binding (e.g., DNA-encoded library technology) may be applied to the discovery of ligands for nanoparticle modification to improve delivery efficiency or effectively induce biological effects (e.g., regulating the protein corona or achieving proximity induction of biomacromolecules).

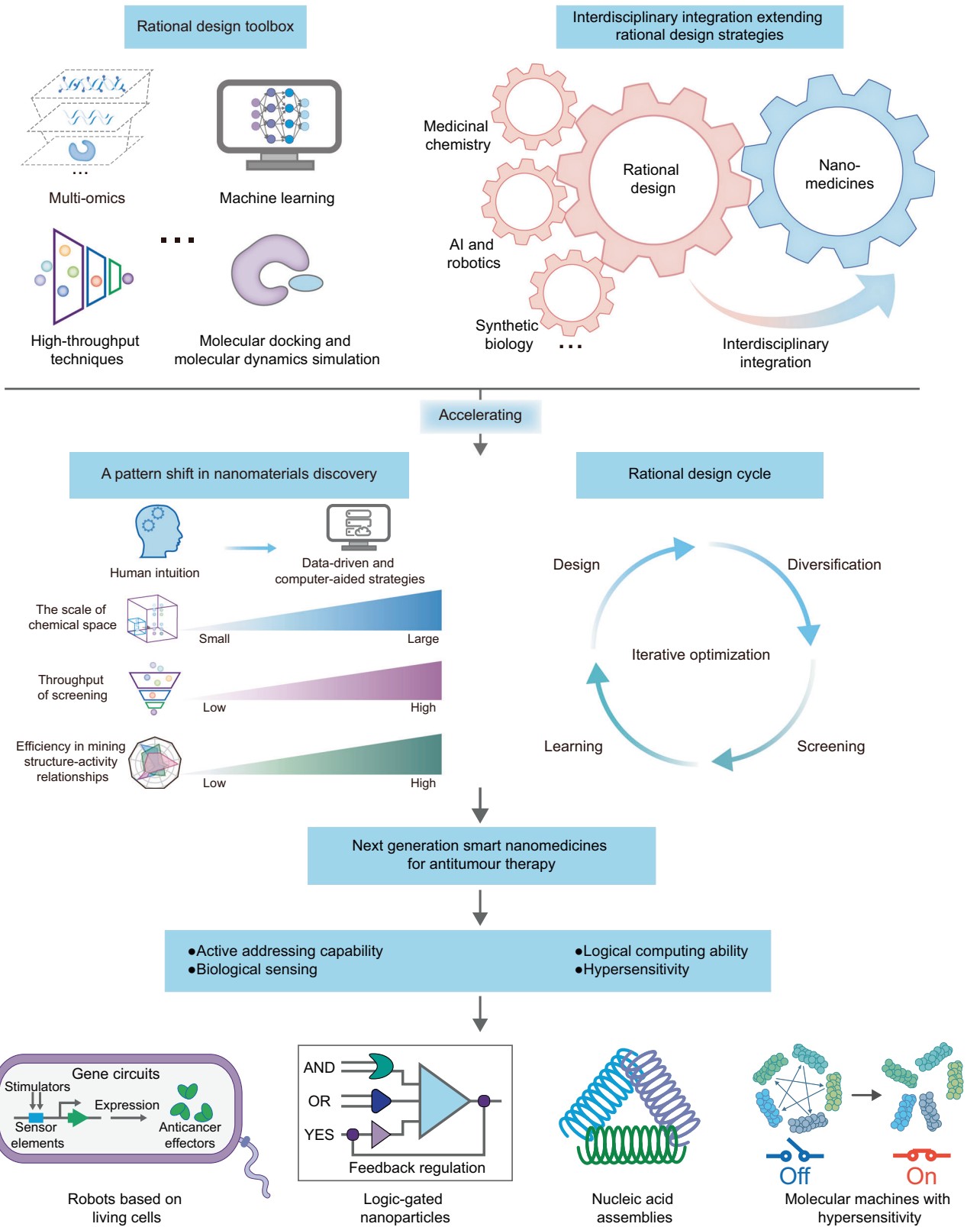

**Fig. 3 | The expansion and application of rational strategies for improving the efficiency of the design and discovery of nanomedicines.** Interdisciplinary integration can enrich rational design and discovery strategies for nanoparticles. Rational strategies are conducive to the directed evolution of nanomedicines to obtain nanoparticles with the desired properties. Rational design strategies have the potential to discover nanomedicines capable of precise drug delivery and controllable release for enhancing antitumour therapy.

The scope of antitumour nanomedicines is expanding, and many carriers with intrinsic therapeutic activities are developing[53], going beyond traditional nanocarriers that deliver chemotherapeutic drugs to tumor cells[54]. Mechanism clarification at molecular levels will contribute to the development of nanomedicines with intrinsic activities, such as nanoparticles with multivalent binding, protein degradation, or membrane rupture abilities[14,53,55–57]. Molecular docking and MD provide powerful tools to reveal the formation mechanisms of nanoparticles and antitumour mechanisms at the molecular level. Molecular docking has been used to explore the immune activation mechanism of lipid nanoparticles and polymers[14,55,56]. Studies have revealed that polyvalency and main chain effects of polymers could increase the interactions between ligands and the stimulator of interferon genes (STING) proteins, thus enhancing immune activation[55,56]. In addition, because of multiple modification sites and multivalence, nanoparticles have the potential to induce proximity to regulate biological processes (e.g., protein degradation) and achieve anticancer effects[58]. Molecular docking and MD have been used to elucidate the proximity induction processes of in situ peptide self-assembled nanoproteolysis targeting chimeras (Nano-PROTACs) degrading epidermal growth factor receptors and androgen receptors (AR)[57]. In addition, MD has been used to study the interactions between nanoparticles and cell membranes to reveal the unique antitumour mechanisms of nanomedicine. Plasma membrane rupture (PMR) strategies are advantageous in inducing cell death because membrane disruption can usually bypass the intracellular signal pathway of target cells and is not affected by drug resistance and metabolic heterogeneity[59]. Based on the feature that cancer cells present higher levels of anionic membrane components than normal cells[59], PMR-based molecules with cationic and hydrophobic modules have been designed and provide a promising strategy for addressing drug resistance in cancer therapy. In one study, the mechanism of membrane rupture of polymers with cationic and hydrophobic parts was revealed by atomic MD. The results illustrated that cation–π interactions promoted the insertion of hydrophobic parts containing benzyl groups into cell membranes, contributing to strong membranolytic activity[53]. Nanomedicines with PMR capability represent a type of nanomaterial that goes beyond the scope of traditional inert excipients and has intrinsic potential therapeutic activities.

## Outlook

Computer-aided rational strategies are expected to facilitate the shift in nanomaterial design and discovery patterns to data-driven modes. By expanding the design space, increasing screening throughput, exploring structure-activity relationships, and elucidating nano-bio interactions, it is possible to improve the design and discovery efficiency of nanomedicines as well as promote iterative optimization (Fig. 3). Rational strategies for improving the efficiency of the design and discovery of nanomedicines can be further expanded and refined in the future through interdisciplinary integration of advanced developments from medicinal chemistry, computer science, synthetic biology, robotics and so on[54,60]. For example, network science provides a powerful tool for addressing the complexity of biomaterials and biological systems, and has been used to solve the complexity of multi-scale materials and open-system material design[61]. Network science is expected to enrich the toolbox for rational design of nanomedicines against cancer. Specifically, identifying key nodes in the networks can promote rational drug combinations[62,63], assist in the screening of materials with interaction forces to achieve self-assembly[26], reveal the mechanism of nano-bio interactions, and predict biomarkers associated with nanoparticles[33]. In addition, network science helps to boost the interpretability of machine learning models for predicting nano-bio interactions[64]. In addition, AI-based structure prediction and generation tools, such as AlphaFold and Rosetta, can efficiently leverage vast structural data in protein databases, providing promising

strategies for the de novo design of biomacromolecular complexes with programmable functions[65] and are expected to be used in the development of nanomedicines. Specifically, the highly symmetrical components created using protein interface design methods enable multivalent presentation of antigens and are expected to be used in the design of nanovaccines[66]. In addition, designing synthetic molecular machines that can control movement or molecular switches that change conformation in response to external stimuli using de novo design tools is expected to achieve efficient drug delivery[65,67]. In addition, with the improvement in computational methods for predicting the binding and formation of biomolecular complexes[68], such as protein-ligand and protein-nucleic complexes, the use of binding principles may accelerate the development of nanocarriers, similar to the binding of paclitaxel and albumin to form nanoparticles. For example, the prediction of the binding and formation of nucleic acid-based complexes has the potential to accelerate the rational development of nucleic acid-based nanocarriers. Complexes generated by de novo design can create material properties that are difficult to achieve by traditional protein engineering or by mimicking natural biological macromolecules[67] and are expected to be used to design next-generation nanomedicines to promote precision cancer therapy. Next-generation nanomedicines (e.g., nanorobots, microrobots, and molecular machines) may have the ability to achieve active addressing, biological sensing, logical computing, feedback regulation, and hypersensitivity[69–71]. Rational design and discovery strategies have the potential to improve the design efficiency of nanomedicines and promote the development of nanomedicines with high drug delivery efficiency.

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

## Acknowledgements

Financial support from National Key R&D Program of China (2022YFC3401404), the National Natural Science Foundation of China (32170935 and 31930066), and the Shanghai Rising-star Program (23QA1411200) are gratefully acknowledged.

## Author contributions

Yaping Li, Dangge Wang, and Xiaoting Shan structured and composed an outline of the manuscript. Xiaoting Shan and Dangge Wang wrote the manuscript and created the figures. Ying Cai, Binyu Zhu, Lingli Zhou, Xujie Sun, Xiaoxuan Xu, and Qi Yin provided extensive feedback on the revision of the initial manuscript and figures. Dangge Wang and Yaping Li revised and supervised the manuscript. All authors contributed to the preparation of the manuscript and approved the final manuscript.

## Competing interests

The authors declare no competing interests.
