## [Transparent Peer Review file · Nature Communications]

Rational strategies for improving the efficiency of design and discovery of nanomedicines

Corresponding Author: Professor Yaping Li

This manuscript has been previously reviewed at another journal. This document only contains reviewer comments, rebuttal and decision letters for versions considered at Nature Communications.

Version 0:

Reviewer comments:

Reviewer #1

(Remarks to the Author)

The authors have addressed the comments raised by reviewer 2. However, in my opinion, there are still a few concerns that should be addressed.

1. In the abstract and the introduction, the authors state that the current workflow for the development of nanomedicines is mainly based in "trial and error". I do not think this is any longer the case, as exemplified by the several rational and high-throughput strategies that these authors discuss in the manuscript. I would avoid such a statement because it does not reflect the current nanomedicine research landscape. Instead, I would highlight that rational and high-throughput design and analysis increasingly widespread in nanomedicine research.

2. There are grammatical errors in the manuscript and clarity needs to be improved in certain sections.

Version 1:

Reviewer comments:

Reviewer #1

(Remarks to the Author)

The authors have satisfactorily addressed my concerns, and I have no further ones

Response to reviewer

Reviewer #1

The authors have addressed the comments raised by reviewer 2. However, in my opinion, there are still a few concerns that should be addressed.

1. In the abstract and the introduction, the authors state that the current workflow for the development of nanomedicines is mainly based in "trial and error". I do not think this is any longer the case, as exemplified by the several rational and high-throughput strategies that these authors discuss in the manuscript. I would avoid such a statement because it does not reflect the current nanomedicine research landscape. Instead, I would highlight that rational and high-throughput design and analysis increasingly widespread in nanomedicine research.

Response: We appreciate the reviewer for helping us to improve the manuscript. We have revised the expressions in the abstract and introduction sections.

(1) In the abstract, we have revised the statement. "The rise of rational strategies in nanomedicine development, such as high-throughput methods and computer-aided techniques, has led to a shift in the design and discovery patterns of nanomedicines from a trial-and-error mode to a rational mode. This transition facilitates the enhancement of efficiency in the preclinical discovery pipeline of nanomaterials, particularly in improving the hit rate of nanomaterials and the optimization efficiency of promising candidates." (Page 2 Line 2-7).

(2) In the introduction, we have introduced the transition of nanomaterial design and discovery mode from a trial-and-error pattern to a computer-aided rational mode by taking the preclinical discovery of lipids as an example. "Rational strategies, such as high-throughput strategies and computer-aided techniques, are increasingly widespread in nanomedicine development to meet the demand for designing nanoparticles efficiently, considering the diversity and complexity of nanomedicines and improving the hit rate of effective candidates in the vast chemical structure space¹.
² The transition of the design pattern from a trial-and-error mode to a rational mode is particularly prominent in the process of lipid discovery and has been gradually

extended to the design of various nanomaterials.” (Page 4 Line 2-8) and “However, discovering and optimizing promising lipid candidates through human-centered trial-and-error strategies is a long and time-consuming process, and some new chemical structures may be difficult to be discovered by human intuition. For example, ionizable lipid DLin-MC3-DMA (MC3), the key component of the first FDA-approved lipid nanoparticle loaded with siRNA (Onpattro), has undergone a long journey of structural optimization (from 2005 to 2012) to improve delivery efficiency and safety⁵⁻⁷. Subsequently, combinatorial chemistry was used for the high-throughput synthesis of lipid libraries to generate structurally diverse libraries for screening effective candidates. However, physical libraries based on combinatorial synthesis often cover limited structural spaces and may miss some promising candidates⁸. Fortunately, the rise of computer-aided high-throughput technologies (e.g., artificial intelligence and virtual screening) is reshaping the process of nanomaterial design and discovery, leading to a shift in the development mode of nanomedicine towards a data-driven and rational pattern. Specifically, computer-aided nanomedicine design and discovery strategies have demonstrated satisfactory potential to expand the structural space of nanoparticle building blocks to improve the likelihood of identifying effective candidates, accelerate the screening process, speed up the mapping of structure-activity relationships, and elucidate nano-biological interactions.” (Page 4 Line 12-28 and Page 5 Line 1-2).

2. There are grammatical errors in the manuscript and clarity needs to be improved in certain sections.

Response: Thank you for the suggestion. We have polished the language in the revised manuscript. Grammatical errors have been addressed in the revised manuscript (e.g., Page 2 Line 11-13, Page 9 Line 16 and Page 11 Line 20). Besides, we have revised the statement to enhance clarity, especially for certain specialized nouns. For example, we have revised "compound libraries" as "excipient libraries" for clarity (Page 9 Line 24).